# Field Evaluation of Low-Cost Particulate Matter Sensors in Beijing

**DOI:** 10.3390/s20164381

**Published:** 2020-08-05

**Authors:** Han Mei, Pengfei Han, Yinan Wang, Ning Zeng, Di Liu, Qixiang Cai, Zhaoze Deng, Yinghong Wang, Yuepeng Pan, Xiao Tang

**Affiliations:** 1State Key Laboratory of Atmospheric Boundary Layer Physics and Atmospheric Chemistry, Institute of Atmospheric Physics, Chinese Academy of Sciences, Beijing 100029, China; meihan@mail.iap.ac.cn (H.M.); wangyinghong@mail.iap.ac.cn (Y.W.); panyuepeng@mail.iap.ac.cn (Y.P.); tangxiao@mail.iap.ac.cn (X.T.); 2State Key Laboratory of Numerical Modeling for Atmospheric Sciences and Geophysical Fluid Dynamics, Institute of Atmospheric Physics, Chinese Academy of Sciences, Beijing 100864, China; liudi@mail.iap.ac.cn (D.L.); caiqixiang@mail.iap.ac.cn (Q.C.); 3College of Earth and Planetary Sciences, University of Chinese Academy of Sciences, Beijing 100049, China; 4Laboratory of Middle Atmosphere and Global Environment Observation, Institute of Atmospheric Physics, Chinese Academy of Sciences, Beijing 100029, China; dengzz@mail.iap.ac.cn; 5Department of Atmospheric and Oceanic Science, and Earth System Science Interdisciplinary Center, University of Maryland, College Park, MD 20742, USA; zeng@umd.edu

**Keywords:** low-cost particulate matter sensors, field evaluation, impact of air humidity, high PM concentration

## Abstract

Numerous particulate matter (PM) sensors with great development potential have emerged. However, whether the current sensors can be used for reliable long-term field monitoring is unclear. This study describes the research and application prospects of low-cost miniaturized sensors in PM_2.5_ monitoring. We evaluated five Plantower PMSA003 sensors deployed in Beijing, China, over 7 months (October 2019 to June 2020). The sensors tracked PM_2.5_ concentrations, which were compared to the measurements at the national control monitoring station of the Ministry of Ecology and Environment (MEE) at the same location. The correlations of the data from the PMSA003 sensors and MEE reference monitors (*R*^2^ = 0.83~0.90) and among the five sensors (*R*^2^ = 0.91~0.98) indicated a high accuracy and intersensor correlation. However, the sensors tended to underestimate high PM_2.5_ concentrations. The relative bias reached −24.82% when the PM_2.5_ concentration was >250 µg/m^3^. Conversely, overestimation and high errors were observed during periods of high relative humidity (RH > 60%). The relative bias reached 14.71% at RH > 75%. The PMSA003 sensors performed poorly during sand and dust storms, especially for the ambient PM_10_ concentration measurements. Overall, this study identified good correlations between PMSA003 sensors and reference monitors. Extreme field environments impact the data quality of low-cost sensors, and future corrections remain necessary.

## 1. Introduction

Aerosols significantly impact global and regional climate change by influencing the physical processes of cloud precipitation, atmospheric radiation, and photoelectric effects [1,2]. The size of aerosol particles mostly ranges from 0.003 to 10 µm, and aerosols can usually be divided by particle size distribution (PSD) into Aitken mode (radius < 0.08 µm), accumulation mode (0.08 µm < radius < 2 µm), and coarse mode (radius > 2 µm) [3]. Aerosols are emitted from both natural and anthropogenic sources, which increases the uncertainty in quantifying the spatiotemporal distribution of aerosols [4]. In recent years, PM_2.5_ (fine particulate matter (PM) with aerodynamic diameters smaller than 2.5 µm) has been widely studied due to its adverse effects on the environment and health issues, including heart disease, lung cancer, decreased cognitive function, and increased mortality [5,6,7]. Fine particles have an important impact on air quality and visibility, and PM_2.5_ has a smaller particle size, larger particle area, and stronger activity than coarser atmospheric particles and easily attaches to toxic and harmful substances (such as heavy metals and microorganisms) [8]. The long residence times and transportation distances of aerosols result in impacts to the quality of the atmospheric environment [5,9]. Many studies have shown that PM_2.5_ has a direct impact on human health: it may cause a variety of diseases and even reduce the average life span [10]. Therefore, continuous observations of PM_2.5_ particles are of great significance.

With the development of microelectronics and other technologies, low-cost optical sensors have been widely applied to monitor the concentrations of particular matter such as PM_2.5_ [11,12,13]. These types of sensors mainly use light-emitting diodes (LEDs) as a near infrared (NIR) optical source and calculate the particle mass concentration by detecting the scattering intensity of aerosol particles in a certain space [13]. To inversely determine the particle mass concentration, Mie scattering theory is used to model the relationship between the scattering intensity and particle radius based on the assumption of spherical particles [14,15]. Through a series of laboratory calibrations, a model was built to determine the mass concentrations of different particle sizes. According to Mie theory, the extinction efficiency and scattering efficiency factors are determined by the wavelength of incident light, the complex refractive index and the radius of aerosol particles, which undergo a nonlinear changing process. For more details, please refer to He et al. [16]. Moreover, low-cost sensors are usually deployed along a short detection light path and are not equipped with environmental control devices, which results in the lower accuracy and precision of these sensors compared with those of research-grade instruments (such as GRIMM). However, many studies have shown that the application of low-cost sensors and high-precision equipment for coordinated observation and the establishment of a calibration model that considers environmental effects such as temperature and humidity can greatly improve the observation accuracy of low-cost sensors [13,17,18]. These studies have made it possible to deploy dense observation networks of low-cost sensors to obtain characteristics of atmospheric PM at high spatial and temporal resolutions.

The accuracy of low-cost sensors is affected by many factors [19]. Therefore, it is important to quantitatively analyze the long-term drift and dynamic span characteristics of this type of sensor. The influence of environmental changes, such as temperature and humidity, must be evaluated simultaneously. Through synthetic laboratory calibration, the aforementioned characteristics of individual sensors can be measured. However, considering the complexity of field observation environments, additional outdoor experiments are required to optimize and verify the accuracy and precision of the calibration model. Additionally, the consistency of different sensors is very important for network observation applications. It is valuable to understand the detection accuracy attributes (such as error distribution) of the sensors in a network for subsequent data applications, such as the error of the observation data and the time-varying characteristics for data assimilation and data fusion.

Ambient PM concentration and meteorological conditions also have a significant impact on sensor accuracy. Rohan et al. [20] quantitatively analyzed the influence of humidity on the performance of several types of low-cost air particle mass sensors and found significant increases in particle number and mass concentrations at relative humidity (RH) values greater than approximately 75%, which suggests that the observations of low-cost sensors cannot be used to determine if air quality standards are being met. Zheng et al. [21] found that PMS3003 sensors (one kind of low-cost sensor) could measure PM_2.5_ concentrations within ~10% of ambient values determined by field evaluations. Zamora et al. [22] suggested that PMSA003 sensors were the most accurate for PM with diameters below 1 μm, while these sensors poorly measured PM in the 2.5–5 μm range. The accuracy of the sensors was dependent on RH, with decreases in accuracy observed at RH > 50%. Ji et al. [23] even developed a simplified method to predict the PM_2.5_ concentrations from measurements of visibility and RH. Therefore, many studies have emphasized and confirmed the effect of humidity on PM measurements.

Assessments of the laboratory and field performance of these low-cost sensors are critical to reduce the uncertainty about the quality of low-cost sensor data. There are limited studies and analyses on the performance of such low-cost sensors and their long-term comparisons in China. Most of the studies on low-cost PM sensors are based on field and laboratory assessments, which were conducted in countries and regions with good air quality (PM_2.5_ concentration < 50 µg/m^3^), such as the United States and Europe [16,22,24,25,26,27,28,29,30,31,32,33,34,35]. Zheng et al. [21] and Gao et al. [36] tested the performance of low-cost sensors in urban environments with high PM concentrations. Zheng et al. [21] conducted Plantower PMS3003 field assessments in Africa (PM_2.5_ concentration < 300 µg/m^3^), but the test time was only slightly longer than one month, so the performance of low-cost PM sensors at high concentrations could not be fully evaluated. Gao et al. [36] compared the performance of seven low-cost PM sensors in Xi’an, China, against the performance of other reference instruments and demonstrated that the low-cost PM sensors could be used to enhance existing PM_2.5_ sampling networks to increase the spatiotemporal resolution of PM_2.5_ datasets. Therefore, long-term field testing of sensors under various concentrations and environments is important for understanding and evaluating the value of low-cost PM sensors [19]. In this study, we evaluated the accuracy and precision of five Plantower PMA003 sensors under realistic conditions, including a range of air humidity conditions, to determine their efficiency when exposed to severe and changing field environments.

## 2. Materials and Methods

### 2.1. Field Deployment

Our experiment was mainly divided into two stages. The first stage was a field comparison between various low-cost, low-precision sensors and the medium-precision Dylos and high-precision GRIMM instruments. The second stage of field testing focused on the performance of the best performing sensor model via a field comparison experiment between multiple sensors of this model and the national control monitoring station of the Ministry of Ecology and Environment (MEE) at the same location. In the first stage, we compared the low-cost sensors with two reference instruments: 1) GRIMM and 2) Dylos instruments. For the first comparison, we directly placed the low-cost sensors at the inlet of the reference instrument. These instruments were deployed for short-term testing at the Xianghe Atmospheric Comprehensive Observation and Experiment Station, which is located in Daluotun, Shuyang, Xianghe County, Hebei Province. To ensure that the instruments were close to the GRIMM standard instrument, the sensors were placed on top of a cabin located at the GRIMM inlet. For the second comparison, we placed a Dylos (See Appendix A) instrument with low-cost sensors in the configuration shown in Figure 1 at the Institute of Atmospheric Physics, Chinese Academy of Sciences (IAP, CAS). Through the comparison of different sensor models in the first stage, we selected the sensor model that performed the best. In the second stage, multiple sensors of the best sensor model were deployed in Beijing for long-term testing starting in October 2019. We still deployed the low-cost sensors at the Institute of Atmospheric Physics, Chinese Academy of Sciences (IAP, CAS) (39°47′ N, 116°57′ E), within 2 km of the Olympic Stadium Center MEE site (39°98′ N, 116°40′ E) (Figure 1), which is a reference site of the national control monitoring station of the MEE. The monitoring site is located in the urban residential area of Beijing, surrounded by roads, so it is a typical traffic site.

### 2.2. Sensor Configuration

First, we compared five types of PM sensors, namely, Plantower PMSA003, Shinyei PPD42NS, NOVA SDS011, and Dylos DC1700 (Table 1), and compared them with the reference instrument GRIMM EDM 180. Figure 2 shows the field instruments. The PM sensors were connected to a Raspberry Pi microprocessor and, later, an updated version of BeagleBone Green Wireless (BBGW) through teletypewriter-universal serial bus (TTY-USB), and meteorological data, i.e., temperature, RH, and air pressure, were collected from the Adafruit BME280 sensor. All data were collected at a temporal resolution of 2 s and stored in the microprocessor using Python scripts and were later uploaded to a remote server through the Internet of Things. The date and time were automatically synchronized from the Alibaba Cloud through a network time protocol daemon (ntpd) service and ensured by a real-time clock (RTC) module in case of poor service.

### 2.3. Evaluation Parameters

For data quality control, we filtered the anomalous observations based on the 3-σ principle every minute. All of the observations were recorded every 2 s and averaged over 1 h. Due to the influence of the AC power supply, some abnormally high values (>1000 μg/m^3^) appeared in the collected raw data. We performed a process to remove extreme values (>1000 μg/m^3^). We used four parameters to evaluate the performance of the PM sensors. The coefficient of determination (*R*^2^) (Equation (1)) was used to reflect the relationship between two sensors. The *R*^2^ results range from 0 to 1, and the larger the *R*^2^ value is, the better the correlation relationship between the two sensors. The root mean square error (*RMSE*) (Equation (2)) was used to evaluate the error between the low-cost sensor and reference instrument, and it gives the highest weight to measurements with the greatest difference to reflect whether the sensor is reliable in heavily polluted conditions. The percentage relative bias (Equation (3)) was calculated to determine the measurement error of the sensors compared with the reference instrument. The percentage relative bias (Equation (4)) was also used to determine the consistency between two sensors.
(1)$$R^{2}={\left[\frac{\sum (sensor_{i}-\bar{sensor_{i}})(sensor_{j}-\bar{sensor_{j}})}{\sqrt{\sum {\left(sensor_{i}-sensor_{i}\right)}^{2}\sum {\left(sensor_{j}-sensor_{j}\right)}^{2}}}\right]}^{2}$$
(2)$$RMSE_{i,j}=\sqrt{\frac{\sum {\left(sensor_{i}-sensor_{j}\right)}^{2}}{n}}$$
(3)$$Relative\;bias_{i}=\left(\frac{sensor_{i}-reference}{average\;\left(reference\right)}\right)*100\%$$
(4)$$Relative\;bias_{i,j}=\left(\frac{sensor_{i}-sensor_{j}}{average\;\left(sensor_{i}\;,sensor_{j}\right)}\right)*100\%$$
where *sensor_i_* and *sensor_j_* are low-cost PM sensor *i* measurements and sensor *j* measurements, respectively, and *n* is the number of sensor *i* measurements.

## 3. Results and Discussion

### 3.1. Comparison of the Cost and Performance of Different PM Sensors

Three types of PM sensors, one medium-precision instrument, and one research-grade reference instrument were used to evaluate the performance of individual PM sensors in this paper. The manufacturer specifications for the Plantower PMSA003, Shinyei PPD42NS, NOVA SDS011, and Dylos DC1700 sensors and GRIMM EDM 180 instrument are listed in Table 1. GRIMM EDM180 is a high-precision instrument that is commonly used worldwide and uses the principle of laser scattering to obtain the concentration of atmospheric PM [37]. Dylos DC1700 is a PM sensor with a moderate cost and accuracy. Due to its good reputation and price point, this sensor was also used as a reference instrument in our research [38]. In Table 1, the prices of the low-cost sensors are approximately one-hundredth that of a high-precision standard instrument, and the physical sizes are much smaller than that of a standard instrument. Therefore, the development of these low-cost miniaturized sensors has made it easy to implement and acquire high-density grid and high-resolution observation data.

We tested the performance of three low-cost PM sensors, Plantower PMSA003, Shinyei PPD42NS, and NOVA SDS011, using Dylos DC1700 and GRIMM EDM 180 as reference instruments. A total of three sets of comparison experiments were established: one experiment was conducted with the GRIMM EDM 180 at Xianghe Station in Hebei Province for one week (8 February 2018 to 15 February 2018), and two were conducted with the Dylos DC1700 at the Institute of Atmospheric Physics in Beijing for 3 weeks (8 February 2018, to 28 February 2018).

For the detection of PM, the largest number of low-cost sensor tests was carried out for optical particle counters [13]. These sensors detect PM by measuring the light scattered by particles, as the optical particle counters can directly count particles according to their size. The measurement principle of Plantower PMSA003, Shinyei PPD42NS, NOVA SDS011, Dylos DC1700, and GRIMM EDM 180 is based on light scattering, that is, the laser irradiation scatters on the PM, the detector receives the scattered light pulse signal, the number of suspended particles with different particle sizes is calculated according to the number and strength of the pulse signal, and then the mass concentration is determined from the collected data. For PMSA003, the manufacturer converted the observed voltage signal to mass concentration and the number of particles for six particle size channels. GRIMM can simultaneously obtain PM measurements in 31 particle size channels. In terms of Shinyei PPD42NS, NOVA SDS011, and Dylos DC1700, the results are not converted into mass concentration, so when we compared the results of multitype sensors, the results were normalized to remove the impact of unit inconsistency. The performances of the different PM sensors are shown in Figure 3. The data collected from the sensors were the original voltage signal, and the final concentration was calculated with the conversion formula. Therefore, the original signals were normalized for uniform comparison and then linearly fitted to the normalized data of sensors and standard instruments. In these experiments, the slope (a), intercept (b), *R*^2^, and *RMSE* were calculated to evaluate the fitting results of the sensors and standard instruments. The fitting results of the three experiments showed that Plantower PMSA003 had the best fitting effect, with *R*^2^ = 0.88~0.97 and *RMSE* = 0.02~0.09 after normalization, compared to Shinyei PPD42NS with *R*^2^ = 0.76~0.86 and *RMSE* = 0.09~0.14, and NOVA SDS011 with *R*^2^ = 0.85~0.89 and *RMSE* = 0.05~0.07. Therefore, we chose Plantower PMSA003 for further evaluation and research.

Both PMSA003 and GRIMM are based on the principle of light scattering but have different emission wavelengths (GRIMM is 660 nm and PMSA003 is approximately 860 nm); however, GRIMM also adopts a higher-stability laser, uses a more precise environmental control device, and is equipped with a more refined air pump to sample the atmosphere. This is why GRIMM is a research-quality instrument. However, low-cost sensors are limited by size and cost, and there is no such control equipment like environmental control devices and air pumps for sampling the atmosphere, so the data quality will be greatly reduced. The basic theory for detecting mass concentration by PMSA003 is based on receiving light scattering intensity. According to Mie theory, the scattering efficiency factor or extinction efficiency factor of particle is determined by the radius of sample particles, the wavelength of incident light, and the complex refractive index (CRI) of aerosol particles as shown in Appendix A. In addition, this process shows a strong nonlinear characteristic. PMSA003 first converts the intensity of scattered light to number concentration per 0.1 L and provides 6 size channels for particle number concentration (>0.3, >0.5, >1.0, >2.5, >5.0, and >10.0 mm). Considering the nonlinear features, the conversion process needs a series of calibrations using standard particles with a known particle size distribution, and He et al. [16] thoroughly studied the transfer function of another type of PMS sensor. Given the number concentration for each bin by PMSA003, the mass concentration of PM_2.5_ and PM_10_ is calculated from number concentration under additional assumptions, such as the average density of particles per bin and the same composition for sampled particles. Overall, the accuracy of mass concentration provided by PMSA003 is determined by many factors: (1) from the perspective of detection, the environment humidity and temperature directly influence the scattered light intensity received by sensors; (2) from the perspective of calibration, the algorithms converting the optical signal to mass concentration determine the accuracy of observed results.

### 3.2. Field Tests of the Precision and Accuracy of the Sensors

#### 3.2.1. PMSA003 and Reference Raw Data

Figure 4 compares the 1 h PM_2.5_ mass concentration at MEE sites in Beijing with the results of 5 nearby sensors. For the sake of clarity, 24 h rolling averaged data were presented. All the measurements were conducted over 7 months, from 25 October 2019 to 10 June 2020, under varying meteorological conditions. The sensors were exposed to temperatures ranging from −10 to +44 °C and RH conditions ranging from approximately 5% to 93% (parameters measured inside the measurement box). The maximum value of the 1 h averaged outputs from MEE was 262 μg/m^3^. During the measurement period, the PM_2.5_ concentration in Beijing was on the high side, reaching 200 µg/m^3^ of heavy pollution as many as seven times, with an average of one or two pollution events per week. Figure 4 indicates that the uncalibrated PMSA003 measurements were basically consistent with the trend of the PM_2.5_ concentration collected by the reference instrument. The concentrations were basically the same in low-concentration areas, and PMSA003 was very sensitive to most of the extreme concentration peaks. However, the performance of the sensor became worse in the high-concentration range; as the difference between the sensor and the reference instrument increased, the consistency among the sensors became worse. In general, the operation of the tested PMSA003 sensor was stable throughout this 7-month study.

#### 3.2.2. Comparison of Deployed PMSA003 and Nearby MEE Stations

Figure 5 shows pairwise comparisons of five field-tested PMSA003 sites and one MEE site, which are at a distance of approximately 2 km. These sensors and MEE data were compared using the parameters provided in the Materials and Methods section. The results showed a good linear relationship between these sensors and the MEE equipment, and the linear slope was basically equal to 1. The statistics in Figure 5 show that the correlations between the different PMSA003 and standard instruments are *R*^2^ = 0.85~0.89 and *RMSE* = 17.10~21.09, with a relative bias of 23.43%~29.59%. In a previous study comparing uncorrected low-cost sensors and standard instruments, T. Sayahi et al. found that the Plantower PMS1003/5003 PM_2.5_ measurements were correlated with hourly tapered element oscillating microbalance (TEOM) measurements (*R*^2^ > 0.87) in two consecutive winters, but in the spring (March–June) and wildfire season (June–October) of 2017, the correlations were poor (*R*^2^ of 0.18–0.32 and 0.48–0.72, respectively) [25]. M. Badura et al. observed a strong linear relationship between TEOM and sensors for 1 h averaged data from Plantower PMSA003 sensors (*R*^2^ approximately 0.83–0.89), Nova Fitness SDS011 units (*R*^2^ approximately 0.79–0.86), and one Winsen ZHO3A unit (*R*^2^ approximate to 0.74–0.81) [28]. Bulot et al. evaluated four PM sensors: Alphasense OPC-N2, Plantower PMS5003, Plantower PMSA003, and Honeywell HPMA115S0. A comparison of these sensors with a nearby background station showed a correlation of 0.61 < R < 0.88 [26]. Compared to uncorrected low-cost sensor evaluations in many other studies, the PMSA003 results are basically in the range of those correlations previously reported. The correlations between the five PMSA003 sensors are *R*^2^ = 0.91~0.98 and *RMSE* = 7.63~28.02, with a relative bias of 12.59%~35.90%, which indicates that PMSA003 offers a high consistency and precision. This high PMS intersensor correlation was also reported in Zheng et al. [21] and T. Sayahi et al. [25].

### 3.3. Performance of PMSA003 at Different Concentration Ranges

According to the conclusion in Section 3.2.2, the five PMSA003 sensors exhibited high intersensor correlations, so the following study integrates the data from the five sensors for analysis. In addition, the data used in Figure 6 exclude the data under the environmental condition of RH > 75%, to avoid the excessive interference of RH signals. Section 3.4 analyzes the impact of RH. To examine the performance of PMSA003 sensors for measuring different ranges of ambient concentrations, the MEE data of PM_2.5_ were divided into six intervals of 0~35, 35~75, 75~115, 115~150, 150~250, and >250 µg/m^3^. The six grade ranges correspond to excellent, good, light pollution, moderate pollution, heavy pollution, and severe pollution, following the pollutant standards index of the Environmental Air Quality Index Table of the MEE. The purpose of this classification standard is to evaluate whether low-cost sensors can accurately reflect the air quality standards in the environment under field conditions, which has a great reference for the application value of low-cost sensors.

Figure 6 is a boxplot of the relative bias between five PMSA003 sensors and MEE data at different concentration ranges. Appendix A is a histogram of the average relative bias and the median relative bias of PMSA003 sensors under different concentrations. The boxplot shows that there are some outliers in the data; the average value may be affected by the outliers, and the reliability is poor. Therefore, the median may be more representative of the general level of bias because it is not affected by the outliers. The statistics shown on the boxplot are recorded in Appendix A. The results show that during the seven-month field test from October 2019 to June 2020 in the Beijing area, 19,147 (82.9%) of the PM_2.5_ hourly concentration data points were distributed from 0 to 75 µg/m^3^, which means that the air quality index (AQI) level of PM_2.5_ was mostly excellent or good. However, there were 3953 data points (17.1%) with data above light pollution (PM_2.5_ > 75 µg/m^3^) and 15 data points that suggested severe pollution (PM_2.5_ > 250 µg/m^3^). This result indicates that the underestimation of a measured concentration increases as the PM_2.5_ concentration increases. When the PM_2.5_ concentration is 0~35 µg/m^3^, the relative bias is approximately 0%, and the average error is 10.84%, indicating overestimation errors for measurement concentrations under a low concentration environment. As the PM_2.5_ concentration increases, the relative bias gradually decreases to below 0%. The median relative biases at 35~75, 75~115, 115~150, 150~250, and >250 µg/m^3^ are −5.79%, −12.60%, −14.54%, −17.61%, and −24.82%, respectively. The corresponding mean relative biases are −5.64%, −10.61%, −11.38%, −14.88%, and −17.09%. The result indicates higher underestimation of the measured concentrations in high-concentration environments.

### 3.4. Impact of Air Humidity on Sensor Performance

The five PMSA003 sensor boxes were all equipped with sensors that measured temperature and humidity to monitor the real-time environmental conditions where the sensors were located. Figure 7 shows the hourly PM_2.5_ and RH data from one of the five PMSA003 sensors (Sensor1). PMSA003 significantly overestimates the concentrations during high-humidity events, while PMSA003 performs better during low-humidity periods, which is basically consistent with the MEE concentration. Therefore, we conducted statistical analysis of the data under different humidity conditions. According to the conclusion in Section 3.3, 82.9% of the data are concentrated from 0 to 75 µg/m^3^; when the concentration is less than 75 µg/m^3^, the relative error is ±10%; and the error increases when the concentration exceeds 75 µg/m^3^. Therefore, when analyzing the influence of RH, it is necessary to control the change in and influence of environmental concentration as much as possible. Therefore, the data used in Figure 8 and Appendix A are conditionally controlled, and the data that were selected for analysis were those when the environmental concentration was <75 µg/m^3^.

Appendix A analyzes the median and average of different RH ranges, which shows a very clear trend that the relative bias between PMSA003 and MEE gradually increases when RH increases. When the environmental RH is 0%~60%, the mean and median of the relative bias are both below 0%, indicating slight underestimation of the measured concentrations under a low RH environment. As the RH increases, the relative bias gradually exceeds 0%. The median relative bias was 7.96% and 14.71% at 60–75% and >75% RH, respectively, and the mean relative bias was 8.55% and 15.02%, respectively. This result indicates overestimation of the measured concentrations and high errors under high-humidity environments (RH > 60%). Many previous studies mentioned and successfully tested the effect of RH on low-cost PM sensors. Rohan et al. [20] quantitatively analyzed the influence of humidity on the performance of several types of low-cost air particle mass sensors and found significant increases in particle number and mass concentrations for RHs above approximately 75%. Zamora et al. [22] suggested that the accuracy of the sensors (PMSA003) was dependent on RH, with decreases in accuracy at RH > 50%. Badura, M et al. [28] observed that a high RH was impacted the Nova SDS011 and Alphasense OPC-N2 devices—clear overestimation of outputs was observed for RHs above 80%. Lu et al. [39] and Zheng et al. [21] also found that the low-cost sensors had high errors for concentrations measured under a higher humidity environment. Therefore, the influence of RH needs to be considered when applying low-cost PM sensors in a high-humidity environment. The use of correction factors for high humidity levels should be advantageous for low-cost PM sensors.

### 3.5. Limitations of PMSA003 Sensors

In this research, we found that PMSA003 performs poorly during sand and dust storms. Although the general occurrence of sand and dust storms and sand days in China has declined significantly in recent years [40], unreasonable human land use has made land desertification increasingly serious, and sand storms formed by natural factors cannot be avoided. Therefore, the monitoring of sand and dust storms is also very important. The main pollutants in dust storms are PM_10_, and the PM_2.5_ concentration will also increase significantly [41]. Most low-cost PM sensors on the market can measure PM of various sizes, including PM_2.5_ and PM_10_. The measurement range of PMSA003 is 0.3–10 μm, so PM_10_ can also be measured. March to June is the season of high sand and dust occurrence in Beijing. For research, the study selected three strong sand and dust events from 10 April 2020 to 10 June 2020. Figure 9 shows the raw PM_2.5_ and PM_10_ data measured by one of the sensors. The averaged values of the PMSA003 measurement and the reference instrument during the three sand and dust events are shown in Figure 9. During these three strong sand and dust events, the PM_2.5_ and PM_10_ measured by PMSA003 were much lower than the MEE reference data. Clearly, the average concentration of PM_10_ measured by PMSA003 was 6–11 µg/m^3^ when MEE reached an average of 130–170 µg/m^3^. The PM_2.5_ measured by PMSA003 was 5–10 µg/m^3^, which was also significantly underestimated when PM_2.5_ reached an average of 16–28 µg/m^3^. In periods without sand and dust events, the concentrations of PM_10_ and PM_2.5_ measured by PMSA003 and MEE are similar. This result shows that PMSA003 has a relatively large measurement error for coarser particles. This finding is consistent with the study by Zamora et al. [22]. Due to the short measurement time and limited sand and dust event monitoring processes, we did not perform statistical analysis on the sand and dust event data.

## 4. Conclusions

This study focuses on testing and evaluating the long-term performance of low-cost light-scattering-based sensors (Plantower PMSA003) in real-world environments. After testing the performance of three low-cost PM sensors, we found that Plantower PMSA003 exhibited the best fitting performance with reference instruments compared to Shinyei PPD42NS and NOVA SDS011.

Regarding the accuracy and precision of PMSA003 sensors, the results show that there is a good linear relationship between these sensors and reference instruments (MEE) and among these sensors. The correlations between different PMSA003 sensors and MEE equipment correspond to *R*^2^ = 0.83~0.09, which shows good consistency between these instruments. The correlations among the five PMSA003 sensors also exhibit high intersensor correlations (*R*^2^ = 0.91~0.98). The evaluation results of PMSA003 is similar to the uncorrected low-cost sensor evaluation results presented in many other studies.

This research was conducted under variable environmental conditions, including variable PM concentrations and meteorological conditions. The performance of the PMSA003 sensors in terms of measuring different ranges of ambient concentrations was examined. The results showed increasing underestimation when the PM_2.5_ concentration increased. When the PM_2.5_ concentration was <35 µg/m^3^, the relative bias was approximately 0%. The median relative bias reached −24.82% when the PM_2.5_ concentration was >250 µg/m^3^, which indicated a greater underestimation of the measured concentration under an environment with a higher PM_2.5_ concentration.

A slight impact of high-RH conditions (>75%) was observed from raw PMSA003 data. PMSA003 significantly overestimated the concentrations during high-humidity events, and the relative bias between PMSA003 and MEE gradually increased with increasing RH. When the environmental RH was 0%~60%, the median of the relative bias was below 0%. As the RH increased, the relative bias gradually exceeded 0%. The median relative bias was 7.9% and 14.7% at 60%–75% and >75% RH, respectively. These results indicate overestimation of the measured concentrations and high errors under high-humidity environments (RH > 60%). In addition, we found that PMSA003 performs poorly during sand and dust events, especially when measuring ambient PM_10_ concentrations.

High-cost and high-precision reference instruments always adopt higher stability laser and are equipped with more precise environmental control devices and more refined air pumps to sample the atmosphere. However, low-cost sensors are limited by size and cost, and there is no such control equipment like an environmental control device, so the data quality will be greatly reduced. This study quantitatively assessed the influence of the field environment (PM concentration, temperature, humidity) on the data quality of low-cost sensors without an environmental control device. Overall, we found that PMSA003 is promising for dense network PM monitoring. However, there are still some problems with the data quality of uncorrected low-cost sensors. The influence of the environment needs to be considered when applying low-cost PM sensors to the field. When applied to the detection of high concentrations of PM_2.5,_ such as in areas with emission hot spots, concentration detection hot spots and concentrations dangerous to human health, low-cost sensors can detect these signals (for example, for use in warning systems) but may underestimate them. If we want to use sensors for scientific research and obtain high-quality data, low-cost sensors cannot directly meet quality requirements and need to be corrected. This study revealed the biases and limitations in using these sensors, and the implications of these results will be beneficial in future corrections of low-cost sensor data and dense network observations.

## Figures and Tables

**Figure 1 sensors-20-04381-f001:**
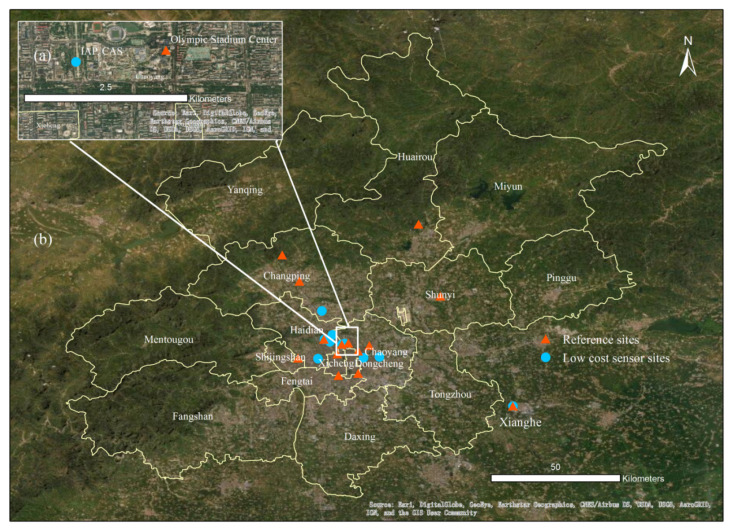
Map of the Beijing and Xianghe deployment sites, showing (**a**) one Beijing site (named Institute of Atmospheric Physics, Chinese Academy of Sciences (IAP, CAS)) and the reference Olympic Stadium Center site and (**b**) the Beijing and Xianghe sites.

**Figure 2 sensors-20-04381-f002:**
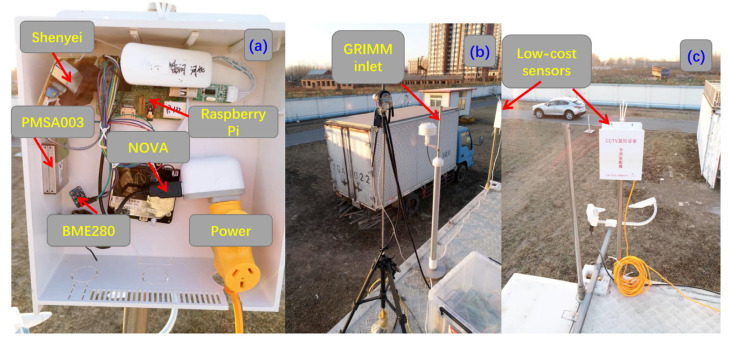
Photographs of low-cost particulate matter (PM) sensors and colocation with a reference instrument (GRIMM) at the Xianghe site. Subplots (**a**), (**b**) and (**c**) are layout of low-cost instrument, inlet of GRIMM and position of low-cost sensors, respectively.

**Figure 3 sensors-20-04381-f003:**
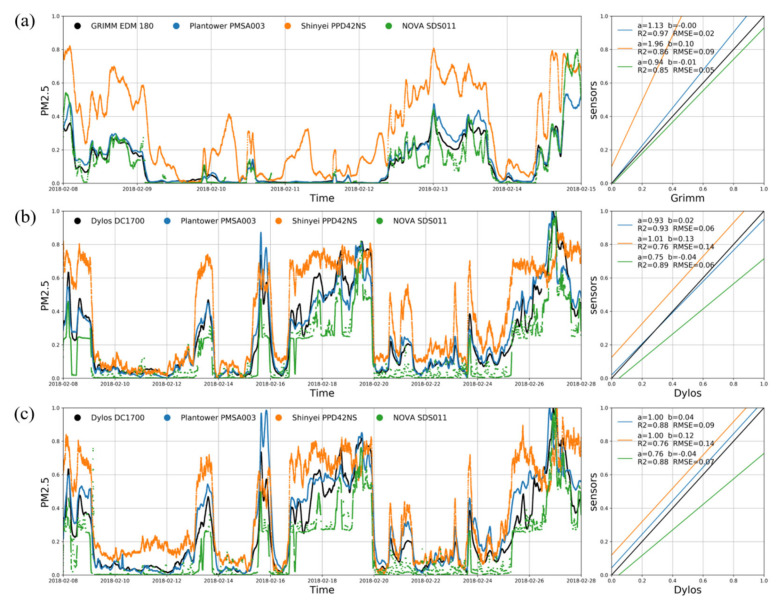
Comparison between (**a**) the first set of PM sensors and GRIMM hourly data from 8 February 2018 to 15 February 2018, (**b**) the second set of PM sensors and Dylos hourly data from 8 February 2018 to 28 February 2018 and (**c**) the third set of PM sensors and Dylos hourly data from 8 February 2018 to 28 February 2018.

**Figure 4 sensors-20-04381-f004:**
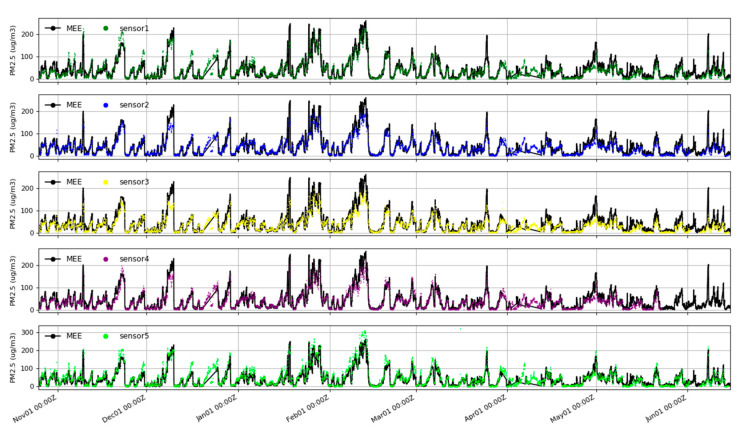
Comparison of PM_2.5_ mass concentrations between the Ministry of Ecology and Environment (MEE) Olympic Sports Center station and the five uncalibrated PMSA003 sensor packages in Beijing from 25 October 2019 to 10 June 2020, based on a 24 h rolling average.

**Figure 5 sensors-20-04381-f005:**
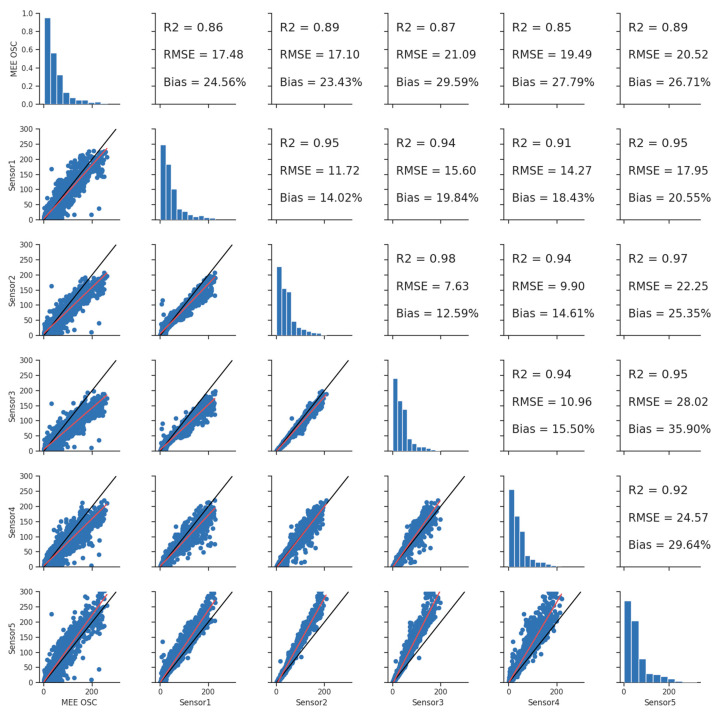
Intercomparisons between five deployed PM sensors and MEE reference instruments at nearby stations. Sensor1–5 refer to the hourly data of the five uncalibrated PMSA003 sensor packages in Beijing from 25 October 2019 to 10 June 2020. The black line is the 1:1 line, and the red line is the fitted line between the two variables in the scatter plot.

**Figure 6 sensors-20-04381-f006:**
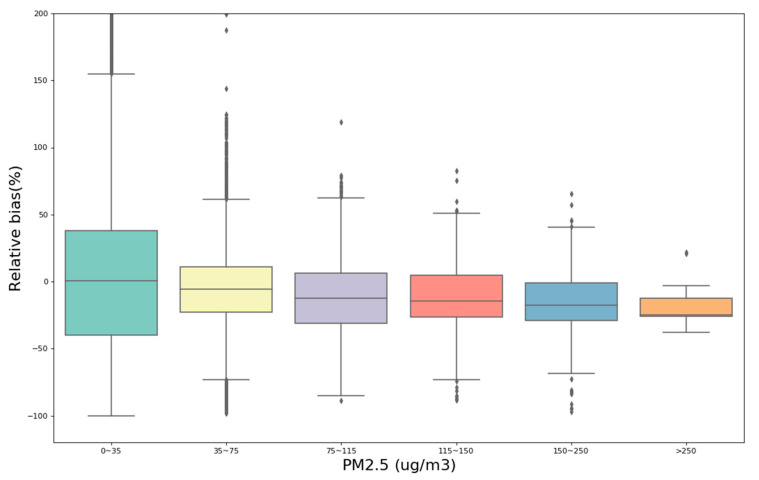
Comparison of the relative bias between five PMSA003 sensors and MEE at different concentration ranges. The values of the lines within the bars represent the medians. The range of the bar indicates the 25th and 75th percentile values (the interquartile range). Whiskers represent values within 1.5 times the interquartile range (IQR), and dots represent values outside this range.

**Figure 7 sensors-20-04381-f007:**
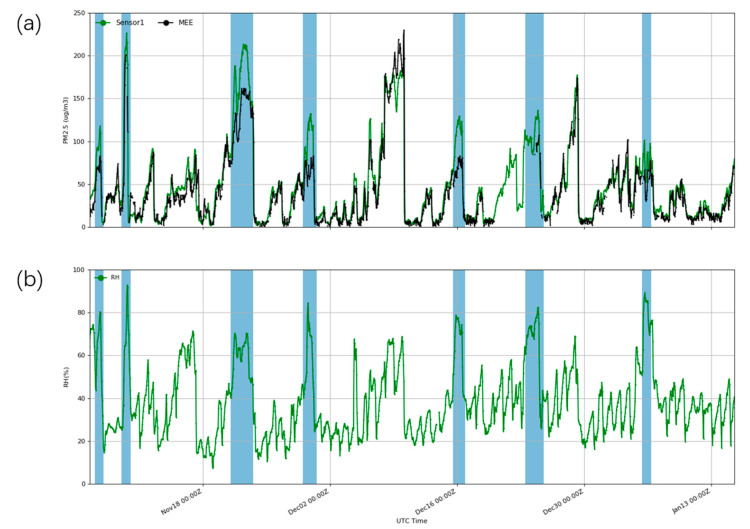
(**a**) Hourly PM_2.5_ concentrations from Sensor1 and MEE from 5 November 2019 to 15 January 2020. (**b**) Hourly RH data from Sensor1. The shaded blue areas indicate high-humidity events (RH > 75%).

**Figure 8 sensors-20-04381-f008:**
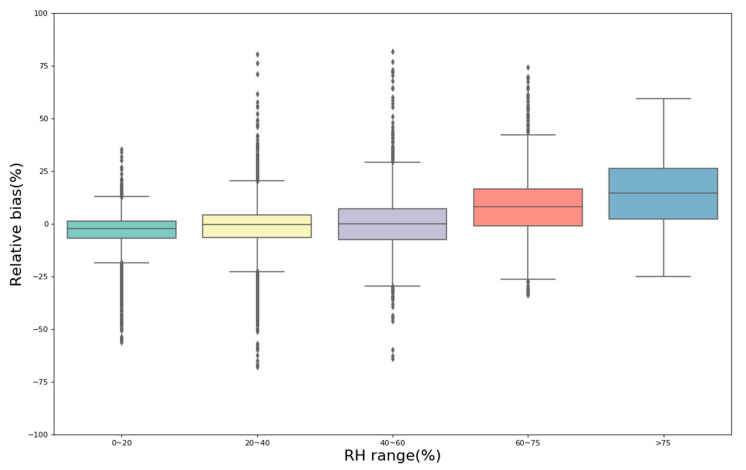
Comparison of the relative bias between five PMSA003 sensors and MEE at different RH ranges based on hourly data with environmental concentrations of 0~75 µg/m^3^.

**Figure 9 sensors-20-04381-f009:**
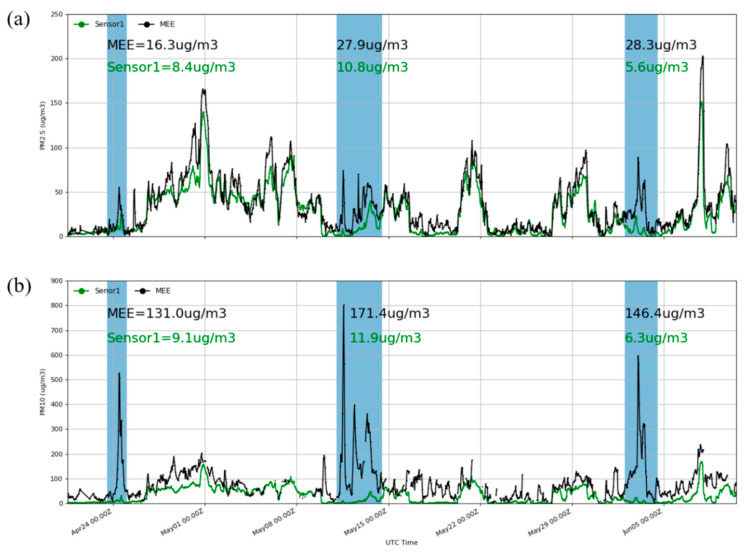
(**a**) Hourly PM_2.5_ concentrations from Sensor1 and MEE from 4 April 2020 to 10 June 2020. (**b**) Hourly PM_10_ concentrations from Sensor1 and MEE during the same time. The shaded blue area indicates a strong dust event.

**Table 1 sensors-20-04381-t001:** Properties of the PM sensors and monitors used in the research.

Parameter	Plantower PMSA003	Shinyei PPD42NS	NOVA SDS011	Dylos DC1700	GRIMM EDM 180
approximate price (¥)	¥ 80.00	¥ 80.00	¥ 130.00	¥ 3800.00	¥ 32,000.00
measuring principle	light scattering	light scattering	light scattering	light scattering	light scattering
range of measurement	0.3–10 μm	>1 μm	0.3–10 μm	>0.5 μm	0.25–32 μm
resolution	1 μg/m^3^	NP *	<0.3 µm/m^3^	NP	NP
manufacturer’s reported precision	±10% @ 100~500 µg/m^3^; ±10 µg/m^3^ @ 0~100 µg/m^3^	NP	±15% (±10 µg/m^3^)	NP	NP
single response time	<1 s	1 s	1 s	NP	6 s
working temperature range	−10 to 60 °C	−30 to 60 °C	−20 to 50 °C	NP	4 to 40 °C
working humidity range	<99% RH	<95% RH	<95% RH	<95% RH	<95% RH
physical size (mm^3^)	38 × 35 × 12	59 × 45 × 22	71 × 70 × 23	190 × 130 × 90	266 × 483 × 364

* NP: Not reported.

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
