# Peer review of "Field Evaluation of Low-Cost Particulate Matter Sensors in Beijing"

_sensors, 2020, doi:10.3390/s20164381_

Round 1
Reviewer 1 Report
In the manuscript ID: sensors-868279, entitled “Field evaluation of low-cost particulate matter sensors in Beijing”, the authors described the evaluation of PM2.5 monitors in Beijing, China, over 7 months. The manuscript is well written, the results well-presented and scientifically sounds and, for this reason, I suggest minor revision before publication. Some comments are reported below.
- Line 28: Please be consistent with PM10
- Line 42: I suggest adding also in this part, the link to the human health.
- Materials and methods: Have the data obtained been treated a posteriori? percentile removal?
- 1 Sensor configuration: I suggest reporting the text before the Figure 1.
- 1 Sensor configuration: I suggest to report the reference to Figure 1 in the text.
- 1 Sensor configuration: I can’t see the “Dylos DC1700” in the Figure 1, as well as the “Adafruit BME280” sensor.
- 2 Field deployment: Also in this case I suggest to report the text before the Figure 2. In general, and regarding all the figures reported in the manuscript, I suggest doing this.
- 2 Field deployment: Perhaps it would not be better to invert the two paragraphs (2.2 and 2.1)?
- 2 Field deployment: In my opinion, it could be useful to add in this section some “field characteristic”. For example, what kind of sites were they? Background sites? Traffic sites?
- 1. Comparison of the cost and performance of different PM sensors: I suggest adding a comparison between miniaturized monitors and reference methods as reported in Borghi et al. 2017 (Int. J. Environ. Res. Public Health 2017, 14, 909; doi:10.3390/ijerph14080909), or as done in the articles reported in the introduction of that review (Velasco et al. 2016; Pokric’ et al. 2015; Mead et al. 2013).
- Figure 3: I suggest improving the quality of the figure and to report it after the reference in the text (line 177).
- Line 176: Can authors also report the monitoring period here?
- Line 197: Temperature and RH have not been evaluated in any other station, outside the measurement box? These values are probably not representative of the real environmental conditions. If so, it should also be better specified in the materials and methods section.
- Figure 5: I suggest to add in the legend what the red and black lines represent.
- Figure 6: How do the authors define “strong dust event”?
- Line 322-324: Please be consistent with subscript and superscript.
- Line 321; Can authors better explain this phrase: “The PM10 measured by PMSA003 was 321 basically 0 μg/m3, and PM2.5 was also significantly underestimated even though the PM10 of MEE reached 322 400-800 μg/m3 and PM2.5 reached 50-100 μg/m3”?
- Supplementary Materials: Are all references to this material listed in the main text?
- Supplementary Materials – tables: I suggest improving the caption of the table (is “std” the standard deviation? 25%?...)
Author Response
We thank the reviewers for understanding the value of this work and valuable comments. We believe the constructive feedback will improve the paper and increase its potential impact to the community. 1. Response to comment: Line 28: Please be consistent with PM10 Thank you for your careful review. Revised accordingly. 2. Response to comment: Line 42: I suggest adding also in this part, the link to the human health. Thank you for your suggestions. We added related description in lines 42-44. In recent years, PM2.5 (particulate matter with aerodynamic diameter smaller than 2.5 µm) as a kind of fine mode aerosol has been widely studied due to its adverse effects on environmental and health, including heart disease, lung cancer, decreased cognitive function and increased mortality[1-3]. 3. Response to comment: Materials and methods: Have the data obtained been treated a posteriori? percentile removal? Thank you. For the data quality control, we had filtered the wild observations based on 3-σ principle every minute. All of the observations were recorded every 2s and averaged into 1 hour values. Due to the influence of the AC power supply, some abnormally high values (>1000μg/m3) may appear in the collected raw data. Our research has done a process to remove the extreme values (>1000μg/m3). High-frequency noise is not a problem for the analysis with the sensor because both datasets are averaged to 1 hour values, which removes most, if not all, of this noise. We added related descriptions in lines 168-171. 4. Response to comment: 1 Sensor configuration: I suggest reporting the text before the Figure 1. Thank you. Revised accordingly. 5. Response to comment: 1 Sensor configuration: I suggest to report the reference to Figure 1 in the text. Thank you. We added the reference to Figure 1 in line 123. 6. Response to comment: 1 Sensor configuration: I can’t see the “Dylos DC1700” in the Figure 1, as well as the “Adafruit BME280” sensor. Thank you. We have added the photos of Dylos DC1700 in figure S1 of the supplement. BME280(Adafruit BME280) was in the lower left corner of Figure 2. Figure 2 is a photo of the Xianghe site, which is located in Daluotun, Shuyang, Xianghe County, Hebei Province. Dylos (Figure S1) were put together with other low-cost sensors in another site in Beijing, so it was not shown in figure 2. 7. Response to comment: 2 Field deployment: Also in this case I suggest to report the text before the Figure 2. In general, and regarding all the figures reported in the manuscript, I suggest doing this. Thank you. Revised accordingly. 8. Response to comment: 2 Field deployment: Perhaps it would not be better to invert the two paragraphs (2.2 and 2.1)? Thank you. Revised accordingly. 9. Response to comment: 2 Field deployment: In my opinion, it could be useful to add in this section some “field characteristic”. For example, what kind of sites were they? Background sites? Traffic sites? Thank you for your suggestion. We added related description in lines 125-126. The monitoring site is located in the urban residential area of Beijing, surrounded by traffic roads, so it is a typical traffic site. 10. Response to comment: 1. Comparison of the cost and performance of different PM sensors: I suggest adding a comparison between miniaturized monitors and reference methods as reported in Borghi et al. 2017 (Int. J. Environ. Res. Public Health 2017, 14, 909; doi:10.3390/ijerph14080909), or as done in the articles reported in the introduction of that review (Velasco et al. 2016; Pokric’ et al. 2015; Mead et al. 2013). Special thanks to you for your good comments. We added measuring principle in table1. And we added related description in lines 218-230. 11. Response to comment: Figure 3: I suggest improving the quality of the figure and to report it after the reference in the text (line 177). Thank you. We adjusted the color scheme, enlarged the font size, and adjusted the locations of legend. 12. Response to comment: Line 176: Can authors also report the monitoring period here? Thank you for your suggestion. We added the monitoring period in lines 215-217. 13. Response to comment: Line 197: Temperature and RH have not been evaluated in any other station, outside the measurement box? These values are probably not representative of the real environmental conditions. If so, it should also be better specified in the materials and methods section. We did not measure the temperature and humidity outside the measurement box, because the purpose of the test is to evaluate the performance of the PM sensors in Beijing under different conditions, so the temperature and humidity in the box are representative of the environment in which the PM sensors are located (i.e., Beijing and surrounding areas). We thank you for your remind and will deploy the instrument to more wide environmental conditions. 14. Response to comment: Figure 5: I suggest to add in the legend what the red and black lines represent. Thank you for your suggestion. We added the description in the legend of figure 5 (lines 298-299). The black line is the 1:1 line and the red line is the fitted line between the two variables in the scatter plot. 15. Response to comment: Figure 6: How do the authors define “strong dust event”? According to "Trial Measures for Early Warning Signals for Sudden Meteorological Disasters" issued by China Meteorology Administration, the event is classified into three degree: 1) Yellow Warning: Dust storms may occur within 24 hours (visibility less than 1000 meters); 2) Orange Warning: Severe dust storms may occur within 12 hours (visibility less than 500 meters) ; 3) Red Warning: Extraordinary sandstorms may occur within 6 hours (visibility less than 50 meters). The definition of strong dust event is based on the sand and dust weather warning as stated above. 16. Response to comment: Line 322-324: Please be consistent with subscript and superscript. Thank you for your careful review. Revised accordingly. 17. Response to comment: Line 321; Can authors better explain this phrase: “The PM10 measured by PMSA003 was 321 basically 0 μg/m3, and PM2.5 was also significantly underestimated even though the PM10 of MEE reached 322 400-800 μg/m3 and PM2.5 reached 50-100 μg/m3”? Thank you. We mean the PMSA003 performed bad for PM measurements in dust weathers, especially for PM10. To quantify this point, we calculated the averaged value of the PMSA003 measurement and the reference instrument during the three sand and dust events and added the value on Figure 9. During these three strong sand and dust events, the PM2.5 and PM10 measured by PMSA003 were much lower than MEE reference data. It can be clearly seen that the average concentration of PM10 measured by PMSA003 was 6-11µg/m3 when MEE reached an average of 130-170 µg/m3. And PM2.5 measured by PMSA003 was 5-10 µg/m3, also significantly underestimated when PM2.5 reached an average of 16-28 µg/m3. We modified related description in lines 400-405. 18. Response to comment: Supplementary Materials: Are all references to this material listed in the main text? Yes, all references to figures and tables in the supplementary materials are listed in the main text, i.e., Figure S1-S3 in Lines 117, 320, 361 and Table S1-S2 in Lines 325 and 359. 19. Response to comment: Supplementary Materials – tables: I suggest improving the caption of the table (is “std” the standard deviation? 25%?...) Thank you for your advice. We have added instructions at the end of the tables. Std: standard deviation; And 25%,50%,75%: 25th ,50th and 75th percentile values (the interquartile range). References 1. Brunekreef, B. and S.T. Holgate, Air pollution and health. Lancet, 2002. 360(9341): p. 1233-1242. 2. Pope, C.A., et al., Lung cancer, cardiopulmonary mortality, and long-term exposure to fine particulate air pollution. Jama-Journal of the American Medical Association, 2002. 287(9): p. 1132-1141. 3. Pope, C.A., III and D.W. Dockery, Health effects of fine particulate air pollution: Lines that connect. Journal of the Air & Waste Management Association, 2006. 56(6): p. 709-742.
Reviewer 2 Report
The authors describe the research and application of low-cost miniaturized sensors in PM2.5 monitoring by comparison with the measurements at the national control monitoring station of the Ministry of Ecology and Environment over 7 months (October 2019-June 2020).
The work presented by the authors is pertinent with respect to the Sensors journal but the results can be presented in a more clear way.
Even if the structure of the paper is ordered, I want to remark the following:
When you introduce an acronym, you have to use capital letters (Ex: Particle Size Distribution, PSD)
Referring to figure 1, can you explain where is located the Dylos DC1700?
Page 4, line 114: a temporal resolution of 2 s is indicated. Can you add more details about why this temporal resolution has been selected? Is it in line with the time response of the selected sensors? At this purpose, as you can see from table 1, the Shinyei pPD42NS sensors has a single response time of 60 seconds.
Page 4, line 114: please add more details about how time and date are acquired.
Page 5, lin 124: why you have chosen 3 km as distance from the MEE sites?
In figure 5, you have to define to what refer “sensor 1”, “sensor 2”, etc.
Page 10, line 214:Even if in the previous page 5 is reported 3 km as a distance from the MEE site, here is reported 2 km. Please revise.
The conclusion section can be improved
Author Response
Reviewer 2 suggested that the results can be presented in a more clear way, we have made corrections in our manuscript according to the Reviewer’s comments.
- Response to comment: When you introduce an acronym, you have to use capital letters (Ex: Particle Size Distribution, PSD)
Thank you for your careful review. Revised accordingly.
- Response to comment: Referring to figure 1, can you explain where is located the Dylos DC1700?
Thank you for your advice. We have added explanation in lines 108-119. We compared the low-cost sensors with two reference instruments: 1) GRIMM and 2) Dylos instruments. For the first one, we directly placed the low-cost sensors at the inlet of the reference instrument. These instruments were deployed for a short-term testing at the Xianghe Atmospheric Comprehensive Observation and Experiment Station, which is located in Daluotun, Shuyang, Xianghe County, Hebei Province. To ensure that the instruments were close to the GRIMM standard instrument, the sensors were placed on top of a cabin located at the GRIMM inlet. For the second one we put Dylos (Figure S1) together with other low-cost sensors same as Figure 1 configuration in the Institute of Atmospheric Physics, Chinese Academy of Sciences (IAP, CAS). But we are sorry that we did not take the photos of Dylos and other co-located PM sensors together as one in IAP site, there is only a detail photo of Dylos (Figure S1), the other sensors were in another box beside same as Figure 1 configuration.
- Response to comment: Page 4, line 114: a temporal resolution of 2 s is indicated. Can you add more details about why this temporal resolution has been selected? Is it in line with the time response of the selected sensors? At this purpose, as you can see from table 1, the Shinyei psPD42NS sensors has a single response time of 60 seconds.
Thank you for your advice. We are very sorry that we have made a mistake on Shinyei's response time. According to the datasheet, 60s refers to the sensor's stabilization time after power-on, and the response time should be 1s. We have revised in Table 1 accordingly. The choice of temporal resolution of 2s is to keep the sampling frequency as high as possible while avoiding potential program error, which also in accordance with other sensors such as BME. And we consider 2s observations allow for resolution of atmospheric variability at short timescales.
- Response to comment: Page 4, line 114: please add more details about how time and date are acquired.
Thank you. We added in lines 142-144. The microprocessor of Rasperry pi and BBGW get date and time from Alibaba Cloud ntpd service; As long as the instrument is connected to the network, the instrument can get network timing from cloud server. There is also a real time clock (RTC) module that uses button battery to ensure date and time is correct for bad networks.
- Response to comment: Page 5, lin 124: why you have chosen 3 km as distance from the MEE sites?
The distance from MEE site is 2 km. We are sorry for the incorrect writing, and we have revised accordingly. Due to the limitation of practical available resources, a place where power and ground can be used. We can only deploy low-cost sensor sites as close to the MEE reference site as possible. Our site is deployed on the roof of the 11-storey building, and there are no obvious emission sources of pollution nearby. Excluding the influence of local emission sources, the aerosols properties within 2 km are similar, and the spatial distribution does not change much, see ref. [1].
- Response to comment: In figure 5, you have to define to what refer “sensor 1”, “sensor 2”, etc.
Thank you for pointing out this. Sensor1-5 refers to the hourly data of the five uncalibrated PMSA003 sensor packages on October 25, 2019, and June 10, 2020, in Beijing. We have added explanation in the legend of figure 5 (line 297-298).
- Response to comment: Page 10, line 214:Even if in the previous page 5 is reported 3 km as a distance from the MEE site, here is reported 2 km. Please revise.
Thank you for your careful review. The distance between low-cost sensor site and the Olympic Stadium Centre MEE site is 2 km, which can be clearly seen from the figure above. We are sorry for the incorrect writing, and we have revised accordingly.
- Response to comment: The conclusion section can be improved.
Thank you for your advice. We have refined the conclusion and added some key summary in lines 420-460.
References
- Lin, C., et al., Using satellite remote sensing data to estimate the high-resolution distribution of ground-level PM2.5. Remote Sensing Of Environment, 2015. 156: p. 117-128.

Reviewer 3 Report
The research and application prospects of low-cost miniaturized sensors in PM2.5 monitoring are described. Five Plantower PMSA003 sensors deployed in Beijing, China, over 7 months (October 2019-June 2020) were investigated. The sensors tracked PM2.5 concentrations, which were compared to the measurements at the national control monitoring station of the Ministry of Ecology and Environment (MEE) at the same location. The correlations between PMSA003 sensors and MEE reference data (R2=0.83~0.90) and among the five sensors (R2=0.91~0.98) showed good accuracy and high correlation between the sensors. However, the sensors tend to underestimate the PM2.5 concentrations at high concentrations. The relative bias reached -24.82% when the PM2.5 concentration was >250 µg/m3. Conversely, overestimation and high errors were found at high relative humidity (RH>60%). The relative bias reached 14.71% at RH >75%. PMSA003 performed poorly during sand and dust exposure, especially for ambient PM10 concentration measurements. Extreme conditions in the field impact the data quality of low-cost sensors, and future corrections remain necessary.
General comments
The paper is well structured and presents the results of a necessary scientific work – sensor QA/QC. But a lot of such studies exists. So, it would necessary to describe the measurement methods of the sensors and instruments in more detail to follow the results of inter-comparison of sensors and instruments and to conclude how the measurement method influences the measurement and inter-comparison results. This is especially necessary because the reference instrument GRIMM EDM180 is working with an optical and not a gravimetric measurement method to determine PM10 and PM2.5 which are units of mass and not particle number.
The application of low-cost sensors provides the question if these sensors can be applied for which tasks:
- Limit values exceedances,
- Hot spot concentration detection,
- Dangerous concentrations for human health,
- Hot spot emission detection.
It would be helpful if these tasks are discussed in the conclusdions.
The paper addresses relevant scientific questions. The paper presents some novel concepts, ideas and tools.
The scientific methods and assumptions are valid and clearly outlined so that substantial conclusions are reached.
The description of experiments and calculations are sufficiently complete and precise to allow their reproduction by fellow scientists.
The quality of the figures is fine.
Title and abstract reflect the whole content of the paper. The abstract should not start with general statements about air pollution.
The overall presentation is well structured and clear. The language must be polished in detail.
The mathematical formulae, symbols, abbreviations, and units are correctly defined.
Specific Comments
Are 7 months a long-term study already because not all seasons of a year are included? The introduction provides the need to study the different weather influences.
Technical corrections
Line 103: efficiency.
It is NOVA SDS011 instead of SDS11.
The measurement unit is required in Fig. 3.
Some citations in the manuscript are missing the reference number.
Author Response
Thanks to the reviewer for the good comments and suggestions. We have made corrections in our manuscript according to the Reviewer’s comments.
General comments
- Response to comment: The paper is well structured and presents the results of a necessary scientific work – sensor QA/QC. But a lot of such studies exists. So, it would necessary to describe the measurement methods of the sensors and instruments in more detail to follow the results of inter-comparison of sensors and instruments and to conclude how the measurement method influences the measurement and inter-comparison results. This is especially necessary because the reference instrument GRIMM EDM180 is working with an optical and not a gravimetric measurement method to determine PM10 and PM2.5 which are units of mass and not particle number.
Thank you. The measurement principle of the Plantower PMSA003, Shinyei PPD42NS, NOVA SDS11SDS011, Dylos DC1700, and GRIMM EDM 180 we use is based on light scattering, that is, the laser irradiation scatters on the particulate matter, the detector receives the scattered light pulse signal, the number of suspended particles with different particle sizes is calculated according to the number and strength of the pulse signal, and then the mass concentration is converted. Both PMSA003 and GRIMM are based on the principle of light scattering but with different emission wavelength (GRIMM is 660 nm and PMSA003 is about 860 nm) , and GRIMM adopts a higher stability laser and uses a more precise environment control device, also equipped with a more refined air pump to sample the atmosphere. That’s the reason of making GRIMM a research-state instrument. As for PMSA003, the manufacturer converted observed voltage signal to mass concentration and the number of particles for six particle size channels, GRIMM can simultaneously get PM measurements in 31 particle size channels. In terms of Shinyei PPD42NS, NOVA SDS11SDS011, Dylos DC1700, the results are not converted into mass concentration, so when we compared the results of multi-type sensors, the results are normalized to remove the impact of unit inconsistency. PMSA003 data’s unit is already the mass concentration in subsequent analysis of the PMSA003, so they can be compared directly with MEE results with the unit of mass concentration. We have added explanations in lines 218-230, and lines 51-67 have tried to introduce how the measurement method influences the measurement accuracy.
- Response to comment: The application of low-cost sensors provides the question if these sensors can be applied for which tasks:
- Limit values exceedances,
- Hot spot concentration detection,
- Dangerous concentrations for human health,
- Hot spot emission detection.
It would be helpful if these tasks are discussed in the conclusions.
Thank you for your good suggestions. We have added discussion in the conclusions (lines 441-447). Overall, our research found that there are still some problems with the data quality of uncorrected low-cost sensors. The influence of environment needs to be considered when applying low-cost PM sensors to the field. When applying to the detection of high concentrations of PM2.5 such as hot spot emissions, hot spot concentration detection and dangerous concentrations for human health, low-cost sensors can detect these signals and give a warning, but may not accurately reflect the true concentration. If we want to use it for scientific research and obtain high-quality data, low-cost sensors cannot directly meet the requirements and need to be corrected.
- Response to comment: The paper addresses relevant scientific questions. The paper presents some novel concepts, ideas and tools.
The scientific methods and assumptions are valid and clearly outlined so that substantial conclusions are reached.
The description of experiments and calculations are sufficiently complete and precise to allow their reproduction by fellow scientists.
The quality of the figures is fine.
The mathematical formulae, symbols, abbreviations, and units are correctly defined.
Thank you for appreciating the value of this work. We will continue to improve in future research.
- Response to comment: Title and abstract reflect the whole content of the paper. The abstract should not start with general statements about air pollution.
Thank you. In the abstract we briefly introduced the research background of low-cost particulate matter sensors and our research significance in one or two sentences at the beginning. Perhaps readers can better understand the purpose and theme of the article. The main part behind is the introduction of the article.
- Response to comment: The overall presentation is well structured and clear. The language must be polished in detail.
Thank you. We have carefully checked the language of the manuscript and have made some modifications.
Specific Comments
- Response to comment: Are 7 months a long-term study already because not all seasons of a year are included? The introduction provides the need to study the different weather influences.
Thank you. What we want to discuss in this study is the impact of ambient temperature, humidity and PM concentration on the sensor measurement. On the time scale of 7 months from October 25, 2019, to June 10, 2020, all measurements were conducted under varying meteorological conditions. Sensors were exposed to temperatures ranging from −10°C to +44°C and RH ranging from approximately 5%–93% (parameters measured inside the measurement box), covering both winter and summer seasons. The maximum value of the 1-h averaged outputs from MEE was 262 μg/m3. The 7-month long-term field environment has adequately covered various environmental changes in the real conditions in Beijing. Moreover, we quite agree with you that the study of seasonal changes is significant, which we will continue following the performances of these sensors and put forward the research.
Technical corrections
- Response to comment: Line 103: efficiency.
It is NOVA SDS011 instead of SDS11.
Thank you for your careful review. Revised accordingly.
- The measurement unit is required in Fig. 3.
Thank you. The data collected from the some sensors are the original voltage signal, and the final concentration needs to be calculated with the conversion formula. As shown in figure 3, the original signals were normalized for uniform comparison and then linearly fitted to the normalized data of sensors and standard instruments. Therefore, the data has no unit after the normalizing process.
- Some citations in the manuscript are missing the reference number.
Thank you. We have carefully checked the manuscript and added relevant references (lines 54, 56, 68, 200, 202).

Round 2
Reviewer 2 Report
Some minor problems persist, please review the document carefully:
Figure 1a is poor: this zoom level may make it possible to add further details.
Line 193: Please review the sensor name "NOVA SDS11SDS011" appears to have been replicated twice.
Row 203: as above
Please review the number of all figures and their reference in the text (there are two figures 1 and 2)
Author Response
Dear Editor and Reviewer:
Thank you for your letter and for the reviewers’ comments concerning our manuscript.
Reviewer 2:
Some minor problems persist, please review the document carefully:
- Response to comment: Figure 1a is poor: this zoom level may make it possible to add further details.
Thank you for your suggestion. We added more details on buildings and road information on Figure 1, indicating the environments of the sensors deployed.
- Response to comment: Line 193: Please review the sensor name "NOVA SDS11SDS011" appears to have been replicated twice.
Row 203: as above
Thank you for your careful review. Revised accordingly in lines 202,213.
- Response to comment: Please review the number of all figures and their reference in the text (there are two figures 1 and 2)
Thank you for your careful review. Revised accordingly in lines 246,268,293,329.

Reviewer 3 Report
The measurement methods of the sensors and instruments were described in more detail but there are no conclusions about the reasons for the described results of inter-comparison of sensors and instruments. So, it is not discussed how the measurement method influences the measurement and inter-comparison results. Further, it is not discussed how the gravimetric units of PM10 and PM2.5 (µg/m3) are determined from the optical measurements results as particle number and particle size distribution and if the different algorithms used are the reason for differences.
This manuscript is more an experiment report than a scientific paper.
Author Response
Dear Editor and Reviewer:
Thank you for your letter and for the reviewers’ comments concerning our manuscript.
Reviewer 3:
- Response to comment: The measurement methods of the sensors and instruments were described in more detail but there are no conclusions about the reasons for the described results of inter-comparison of sensors and instruments. So, it is not discussed how the measurement method influences the measurement and inter-comparison results.
Thank you for pointing out this important issue. We added the reasons for this in lines 224-229. And in the conclusion we described it in lines 427-432.
The measurement principles of Plantower PMSA003, Shinyei PPD42NS, NOVA SDS011, Dylos DC1700, and GRIMM EDM 180 we use are all based on light scattering. Both low-cost sensors and GRIMM are based on the principle of light scattering but have different emission wavelengths (GRIMM is 660 nm and PMSA003 is approximately 860 nm); however, GRIMM also adopts a higher stability laser, uses a more precise environment control device, and is equipped with a more refined air pump to sample the atmosphere. This is why GRIMM is a research-quality instrument. However, low-cost sensors are limited by size and cost, and there is no such control equipment like environment control device and air pump for sampling the atmosphere, so the data quality will be greatly reduced.
That’s the reason for the results of inter-comparison between the sensors and the reference instruments. Our research focused on quantitatively assessment of the influence of the field environment (PM concentration, temperature, humidity) on the data quality of low-cost sensors without environment control device.
- Response to comment: Further, it is not discussed how the gravimetric units of PM10 and PM2.5 (µg/m3) are determined from the optical measurements results as particle number and particle size distribution and if the different algorithms used are the reason for differences.
Special thanks to you for your constructive and invaluable suggestions. We added the discussions on how the gravimetric units of PM2.5 and PM10 (μg/m3) are determined from the optical measurements results as particle number and particle size distribution in lines 229-243, and added Figure S2 to explain this in the Supplementary Materials.
The basic theory for detecting mass concentration by PMSA003 is based on receiving light scattering intensity. According to Mie theory, the scattering efficiency factor or extinction efficiency factor of particle is determined by the radius of sample particles, the wavelength of incident light and the Complex Refractive Index (CRI) of aerosol particles like Figure S2 below showing. And this process shows a strong nonlinear characteristic. PMSA003 converts the intensity of scattered light to number concentration per 0.1 liter first, and provides 6 size channels for particle number concentration (>0.3, >0.5, >1.0, >2.5, >5.0, and >10.0mm). Considering the nonlinear features, the conversion process need series of calibrations using standard particles with a known particle size distribution, and He et al.[1] thoroughly studied the transfer function of another type of PMS sensor. As having the number concentration for each bins by PMSA003, the mass concentration of PM2.5 and PM10 is calculated from number concentration under additional assumptions such as the average density of particles per bin and the same composition for sampled particles. Overall, the accuracy of mass concentration provided by PMSA003 is determined by many factors: (1)From detection view, the environment humidity and temperature directly influence the scattered light intensity received by sensors; (2) From calibration view, the algorithms converting the optical signal to mass concentration determine the accuracy of observed results.
The manufacturers of these low-cost sensors generally do not disclose the algorithms of converting observed voltage signal to mass concentration, which are technically confidential, and we only got the conversion method of PMSA003, so we are sorry that we can not explain if the different algorithms used are the reasons for differences. However, the technology and design of different sensor from different manufactory will definitely be different, which may mainly result in differences in measurement results. The focus of the manuscript is to invest the influence of the field environment on low-cost sensors, which is also the biggest disadvantage of all low-cost sensors. The purpose of testing different sensor models is to find the best one for subsequent analysis, so there is not much discussion in this part.
Figure S2 Aerosol extinction efficiency factor calculated by Mie theory. The x axis is scale factor equal to 2πr/λ,m is complex refractive index. The y axis is aerosol extinction efficiency factor.
- Response to comment: This manuscript is more an experiment report than a scientific paper.
Thank you for the comment. In this paper, we evaluated four types of low-cost sensors (LCS) and found that PMSA003 performed best. We further analyzed temperature, humidity, high PM concentrations, and sand and dust impacts on this type of sensor and revealed the biases and limitations in using it, the implications of these results are beneficial in future corrections of LCS data and dense network observations. We explained our research significance at the end of the conclusion (lines 440-441). We hope this can satisfy the reviewer’s concerns.
Reference:
- He, M., N. Kuerbanjiang, and S. Dhaniyala, Performance characteristics of the low-cost Plantower PMS optical sensor. Aerosol Science and Technology, 2019. 54(2): p. 232-241.
